# Improving Visual Working Memory with Cholinergic Deep Brain Stimulation

**DOI:** 10.3390/brainsci13060917

**Published:** 2023-06-06

**Authors:** Janki M. Bava, Zhengyang Wang, Sarah K. Bick, Dario J. Englot, Christos Constantinidis

**Affiliations:** 1Department of Biomedical Engineering, Vanderbilt University, Nashville, TN 37235, USA; janki.m.bava@vanderbilt.edu (J.M.B.); dario.englot@vumc.org (D.J.E.); 2Neuroscience Program, Vanderbilt University, Nashville, TN 37235, USA; zhengyang.wang@vanderbilt.edu; 3Department of Neurological Surgery, Vanderbilt University Medical Center, Nashville, TN 37232, USA; sarah.bick@vumc.org; 4Department of Ophthalmology and Visual Sciences, Vanderbilt University Medical Center, Nashville, TN 37232, USA

**Keywords:** visual working memory, deep brain stimulation, cholinergic activation

## Abstract

Acetylcholine is a critical modulatory neurotransmitter for cognitive function. Cholinergic drugs improve cognitive performance and enhance neuronal activity in the sensory and association cortices. An alternative means of improving cognitive function is through the use of deep brain stimulation. Prior animal studies have demonstrated that stimulation of the nucleus basalis of Meynert through DBS improves cognitive performance on a visual working memory task to the same degree as cholinesterase inhibitors. Additionally, unlike current pharmacological treatments for neurocognitive disorders, DBS does not lose efficacy over time and adverse effects are rare. These findings suggest that DBS may be a promising alternative for treating cognitive impairments in neurodegenerative disorders such as Alzheimer’s disease. Thus, further research and human trials should be considered to assess the potential of DBS as a therapeutic treatment for these disorders.

## 1. Introduction

Working memory is a core cognitive function that allows us to maintain information in our mind over a period of seconds and underpins more complex abilities such as planning, language, and fluid intelligence [1]. Due to this central role in cognition, improving working memory has been a goal of interventions aiming to restore cognitive function impairments due to a multitude of pathological conditions such as traumatic brain injury, stroke, schizophrenia, and Alzheimer’s disease [2,3,4]. Ascending cholinergic input to the neocortex is a critical neuromodulator in working memory [5]. Cholinergic drugs are frontline medications for many conditions that compromise cognitive function [6]. Visual working memory performance is enhanced through such cholinergic modulation [7]. In recent years, deep brain stimulation (DBS) targeting the cholinergic basal forebrain has emerged as an alternative to pharmacologic approaches to modulate acetylcholine release and improve cognitive function [8]. To understand the promises and challenges involving this approach, here we will review the actions of neuromodulation systems and acetylcholine in particular, and we will discuss the role of cholinergic innervation with respect to visual working memory, the mechanisms of action of deep brain stimulation targeting the cholinergic system that support its viability as an intervention, and the experimental evidence of its effectiveness. As this field is still relatively new, most of the latter evidence comes from animal studies, particularly in non-human primates; this will be the emphasis of our review. 

Monoamines and acetylcholine are important neuromodulatory systems that govern the cognitive functions mediated by the prefrontal cortex (PFC). The interplay between these systems is crucial for the execution of cognitive functions including arousal systems [9]. Fluctuations in dopamine, serotonin, and norepinephrine levels affect the performance of PFC cognitive processes such as working memory, attention, and decision-making. 

In the dorsolateral PFC (dlPFC), neuromodulation of the monoamines and acetylcholine systems are especially critical [9]. Dopamine primarily acts on D1 receptors to increase cAMP production in the primate dlPFC. However, this action is dose dependent and there is a range of optimal stimulation, as too little or too much has been shown to impair working memory function [10]. The administration of D1R agonists has been shown to reduce noise during working memory representations [11]. The increased cAMP production opens K+ channels maintaining delay-related firing of relevant inputs while suppressing the response to distractor inputs [11].

Misfiring and loss of regulation of these neuromodulatory systems in the dlPFC leads to cognitive impairments and is generally seen with aging [9]. While the exact functions of these neuromodulators may not be completely understood, we do know that acetylcholine and dopamine are essential for proper cognitive processing, specifically working memory [12]. Studies focused on monoaminergic and cholinergic innervation and activation will be the key to understanding how we can retain and improve working memory function.

## 2. Cholinergic System

The primate neocortex receives its acetylcholine innervation from the nucleus basalis of Meynert (NB) [13,14]. Nucleus basalis activity levels during wakefulness reflect alertness levels [15] and are systematically related to executive function [16,17,18,19]. Ascending cholinergic projections spread across the neocortex, allowing the system to control the excitability of most cortical areas; however, the modulation can still be specific for particular areas and stimuli [20].

Endogenously released acetylcholine acts on cholinergic receptors, both nicotinic and muscarinic. Nicotinic receptors are ion channels that allow for the passage of calcium and sodium ions into the postsynaptic neuron when acetylcholine molecules are bound to them. This regulation of calcium signaling increases synaptic transmission and plasticity and enhances the release of dopamine. Muscarinic receptors are G-protein coupled receptors that activate the phosphoinositide and cAMP cascading pathways. These activated pathways result in changes to the postsynaptic neuron’s excitability. Specifically, the activation of M1 receptors within the PFC has been shown to increase the signal-to-noise ratio and filter out irrelevant information [9]. Activation of nicotinic α7 receptors similarly facilitates spatial working memory. When acetylcholine binds to the α7 receptors, it allows for the activation of the NMDA receptors, which is crucial for maintaining persistent activity during spatial working memory tasks [12]. The human PFC releases acetylcholine in response to a target in an inverted U-shaped relationship with task performance [21]. In monkey dlPFC neurons, cholinergic agents have reduced the effect of distractors [22], which is consistent with cholinesterase inhibitors enhancing target detection amongst distractors [23]. These results suggest acetylcholine may be essential for perceptual representations in natural environments, among competing stimuli.

Cholinergic action is essential for cognitive functions and behavioral outputs; conditions that cause cholinergic deficits result in cognitive function decline [24,25]. Basal forebrain lesions in non-human primates cause deficits in cognitive domains including spatial attention, working memory, and task acquisition [26,27,28]. Deafferentation of cholinergic projections to the prefrontal cortex impairs monkeys’ selectively in spatial working memory tasks, while leaving performance in tests of decision-making and episodic memory unaffected [29]. Decrease in the action of acetylcholine, for example by administration of muscarinic receptor antagonists, impairs cognitive abilities in both humans and animals [30,31,32,33,34]. Similarly, atrophy of the NB has been observed in cognitive disorders and has been linked to the decline of cholinergic neurotransmission and cognitive impairment [35]. Dysfunction of acetylcholine-releasing neurons in the basal forebrain contributes to the cognitive declines observed in Alzheimer’s disease [17,18,19,36]. Reduced acetylcholine release, alterations in choline transport, and changes in nicotinic and muscarinic receptor expression have been observed [18,19,36].

Conversely, systemic administration of acetylcholinesterase inhibitors is widely used to treat executive function deficits in human patients [37,38,39,40,41]. Such drugs have also been shown to improve performance in working memory tasks in monkeys [18,42]. Acetylcholinesterase inhibitors that increase cholinergic neurotransmission in the brain by preventing the breakdown of acetylcholine can partially reverse cognitive deficits [43,44]. However, the value of cholinergic medications is limited; high doses result in cognitive improvements that peak in ~3 months [45], and typically attenuate after a year. Even moderate doses are discontinued in some patients due to peripheral side effects that arise due to off-target effects of the drugs [44]. 

## 3. Neural Activity Underlying Visual Working Memory

The neural mechanisms underlying visual working memory have been extensively studied in recent years and have been a matter of some debate [46,47]. Improved visual working memory performance involves activation in the dorsolateral prefrontal cortex and the inferior parietal lobule [48,49]. It is notable that similar effects are observed with the administration of a cholinesterase inhibitor during visual working memory tasks in Alzheimer’s patients, including heightened activation in the prefrontal and parietal cortex during the memory maintenance period [50].

Individual neurons exhibit persistent activity with selectivity for spatial locations [51,52]. Computational models typically simulate persistent activity by networks of neurons with recurrent connections between similarly tuned units, as illustrated in Figure 1A [53,54].

Activation in the network behaves as a continuous attractor [56]. An initial stimulus appearance generates a bump (peak) of activity in the network, which is maintained in the delay period, but may drift randomly. The peak of the bump at the end of the delay period determines the location recalled by the subject (Figure 1B), hence the term bump attractor [57]. Drifts in neuronal activity account for deviations of behavior: persistent activity recorded from trials in which monkeys make eye movements deviating clockwise vs. counterclockwise relative to the true location of the stimulus yields slightly different tuning curves (Figure 1, bottom), implying that the peak of activity at the end of the delay period determines the recalled location [57].

PFC neurons also generate persistent discharges in memory tasks that require the maintenance of stimulus features such as shape, color, or direction of motion, and the identity of objects, faces, and abstract pictures [58].

## 4. Action of Acetylcholine on the Visual Cortex

Cholinergic release, during states of elevated alertness or focused attention, enhances the processing of visual information [59]. Acetylcholine controls the flow of sensory information in the cortex by increasing the gain of thalamic input and thus enhances contrast sensitivity, a mechanism mediated by nicotinic receptors [60]. Administration of acetylcholine or cholinergic agonists in the sensory cortices similarly enhances responses to stimuli. This is the case in monkey V1, where responses to stimulus presentations improve under acetylcholine iontophoresis [61]. The effect is blocked by muscarinic antagonists. Similarly, cholinergic agonists in the middle temporal area (MT), an area of the visual cortex implicated in motion perception, affect responses to visual stimuli, which are also blocked by muscarinic antagonists [62]. Importantly, neurons that are not driven by the sensory inputs are suppressed rather than facilitated by acetylcholine administration, thus increasing signal-to-noise ratio as well as reducing top-down influences on sensory responses, such as that related to expectation [21,63]. More direct evidence about the short-term action of acetylcholine has been obtained by optogenetic, phasic stimulation of cholinergic neurons in rodents, which has been shown to increase visual cortical firing rate during the period of visual stimulus presentations [64]. Substantial species differences between primates and rodents have been reported, however [65].

The effects of acetylcholine are not uniform between cortical areas. Whereas in area V1 acetylcholine enhances responses to attended over unattended stimuli [61], no such effect was observed in area MT [62]. The diversity of cholinergic effects can be attributed, at least in part, to different patterns of expression of muscarinic and nicotinic receptors in pyramidal neurons and interneurons in different cortical areas [12,66].

## 5. Effects of Acetylcholine on the Prefrontal Cortex

Cholinergic agonists generally increase the activity of prefrontal neurons [22,67,68]. Conversely, systemic administration of the muscarinic antagonist scopolamine reduces prefrontal activity [69], as does micro-iontophoresis of muscarinic and nicotinic-α7 inhibitors [67,70,71]. The increase in activity by agonists is selective for the preferred location of the neuron, so that tuning is enhanced (schematically shown in Figure 2A). 

This sharpening of prefrontal neurons’ tuning functions takes place with low doses of cholinergic agonists. Broader tuning (as in Figure 2B) has been observed in pharmacological studies by cholinergic overstimulation with high doses of carbachol or M1R allosteric inhibitors administered with iontophoresis in working memory tasks [71,72,73]. 

The decrease in selectivity by cholinergic overstimulation [71,72,73] has been assumed to correspond to the descending section of an inverted-U function, representing a regime over which cholinergic agonists impair performance (as in Figure 2C) [74]. The effectiveness of drugs targeting other neurotransmitter systems, e.g., dopamine, is also often interpreted in terms of increased stimulus selectivity [75]. 

## 6. Deep Brain Stimulation

An alternative method of improving cholinergic function is to selectively activate the NB, the exclusive source of cholinergic innervation of the neocortex in primates, including humans [76]. Such a method has several advantages, including avoidance of side effects caused by peripheral cholinergic targets and control of the specific time intervals during which NB stimulation is applied, e.g., during waking hours, when acetylcholine is normally released [77]. Additionally, stimulation is likely to enable the activation of non-cholinergic projection neurons [78,79,80,81], and the co-transmission of GABA from cholinergic terminals [82,83], better mirroring the physiological conditions of natural acetylcholine release. 

The biophysical mechanisms of DBS are the subject of some debate, but continuous high-frequency stimulation generally used to treat movement disorders appears to overall suppress the activity of the implanted area through overstimulation, by a rate of efferent targets [84,85,86]. This results in a reduced impact of the implanted region on its target areas. In most DBS applications, rectangular monophasic or biphasic voltage pulses are used to evoke the desired physiological effects [87]. Biphasic stimulation consisting of a cathodal stimulation phase followed by an anodic reversal phase is typically used in clinical applications [88,89]. Deep brain stimulation is not cell type specific, as electrical stimulation induces extracellular depolarization of all excitable structures in the target spatial region including fibers of passage irrespective of biochemical composition and morphology, but it is spatially selective around the electrode lead. The activating function, or ability to elicit an all-or-none action potential response, drops off sharply at approximately 2 mm from the electrode lead for a typical current near 3 mA. The square root of the current scales with the radius of the activating function [90,91,92]. The spatial specificity of these fields is quite difficult to match with non-invasive stimulation [93]. Increasingly sophisticated stimulation has been made possible in movement disorders, for example addressing different subtypes of Parkinson’s disease with precise targeting of the subthalamic nucleus or internal segment of the globus pallidus [94].

## 7. Cholinergic Deep Brain Stimulation in Visual Working Memory

A series of recent studies have investigated the use of cholinergic DBS for visual working memory and attention. In one study, monkeys were trained on a delayed-match-to-sample task in which they were required to remember an initial stimulus for a short delay period and then indicate, out of two options, which object was the match of the initial stimulus. Unexpectedly, applying continuous stimulation to the nucleus basalis of Meynert at high frequencies (e.g., 80 Hz or higher) similar to the high frequencies used in movement disorders did not improve but instead impaired behavioral performance in the cognitive task used [95]. After testing different stimulation parameters, an alternative stimulation regime involving intermittent stimulation was found to improve performance. This regime involved delivering 1200 pulses per minute for 15–20 s (i.e., with a frequency of stimulation of 80–60 Hz, respectively) alternating with intervals of no pulses lasting 40–45 s. Moreover, the intermittent stimulation parameters were tested in conjunction with Donepezil, a cholinesterase inhibitor. The combination of DBS with Donepezil did not increase performance any more than DBS alone or Donepezil alone, suggesting a ceiling effect. Further results from this study suggested intermittent stimulation could be blocked by both nicotinic and muscarinic antagonists [95]. The benefits of cholinergic deep brain stimulation were not limited to a single task. In a different study, monkeys receiving cholinergic stimulation performed better in a continuous visual attention task by increasing the hit rate and decreasing the false alarm rate [96]. More recently, cholinergic stimulation has shown promise in improving cognitive function in aged monkeys [97]. 

Another recent study also examined the effects of cholinergic deep brain stimulation on neuronal activity [98]. The pattern of intermittent stimulation allows for recordings to be interleaved with periods of stimulation (Figure 3). 

Neural activity in the prefrontal cortex was recorded during a task in which two stimuli were presented in a sequence. Depending on the color of the fixation point, the monkeys had to make an eye movement to the remembered location of the first or second stimulus. Stimulation of 80 Hz frequency was applied between trials for 15 s, and this was repeated after the proceeding 45 s trial was completed. NB stimulation increased overall activity and behavioral performance during the working memory task (Figure 4).

Unexpectedly, and unlike the results of cholinergic agonists reviewed above, the tuning of prefrontal neurons generally broadened, i.e., neurons responded more strongly to stimuli that were less optimal. Sharper tuning is typically associated with increased performance for conditions in which decoding precise stimulus information is essential, as in discrimination tasks, after perceptual learning [99,100,101,102]. Theoretical studies have shown, however, that broader tuning curves can produce either worse or better performance depending on the task and noise conditions in the network [103,104,105,106]. For conditions in which stimuli are highly discriminable, and performance depends on the ability to filter distractors or other intervening noise and implement the correct rule, as was the case in the task we used, broader tuning may lead to more efficient coding [107]. Unlike the effects of perceptual learning mentioned above, learning to perform a working memory task also induces broadening of neural selectivity [108,109].

Cholinergic NB stimulation had a number of other effects on neural activity. It reduced the trial-to-trial variability of the neuronal firing rate, consistent with the effects of cholinergic drugs [110,111]. NB stimulation also produced changes in the rhythmicity of local field potentials, leading most significantly to a decrease in α power in the prefrontal cortex [112]. 

## 8. Potential for Human Intervention

In a few recent studies (Table 1), human patients were implanted with electrodes that allowed stimulation of the nucleus basalis of Meynert, among other targets [76,113,114,115,116,117,118,119]. Implantation and stimulation pulses in the range of 20 Hz (and higher) were safe and well tolerated, and adverse effects were comparable to those of movement disorders. Improvement in glucose utilization, and some cognitive benefits as captured by the neuropsychiatric inventory, were reported. However, compelling cognitive improvements were not observed for the stimulation group. All of these clinical studies relied on the application of continuous stimulation at 20 Hz or higher, leaving open the possibility that a protocol of intermittent stimulation might be more effective. Animal studies have shown the benefits and cognitive improvements of intermittent stimulation over continuous stimulation, which has not yet been explored in human trials [95]. Using intermittent stimulation allows for greater flexibility of stimulation parameters and reduces the risk of desensitization of the target areas. Furthermore, intermittent stimulation provides a more controlled approach that could be leveraged for more sustainable cognitive improvement. 

There are still challenges and limitations that need to be addressed before DBS can be considered as a therapy for memory deficits, including determination of optimal stimulation parameters. No studies have determined the optimal stimulation protocol for humans. At present, determination of optimal parameters is based on a trial-and-error method that will be a time-consuming process. Furthermore, we must examine the progression of these neurodegenerative diseases to determine the time frame in which intervention will be the most effective. As with any major surgery, implantation of DBS electrodes poses risk. Despite the lack of any reported major adverse effects, implantation surgery may lead to intracerebral hemorrhage, seizures, and other complications, including the risk of infection at the surgical site. There is also the matter of electrode placement. There needs to be a method to precisely determine and implant the electrodes in the most favorable location to ensure the validity of any observed effects and reduce the chances of lead migration and fracture. Although these limitations preclude the wide adoption of the method at present, ongoing research effects may make this a reality.

## 9. Conclusions

The cholinergic system is essential in regulating cognitive processes, and degeneration of the cholinergic system is a key component of neurocognitive disorders. Currently, there are no curative treatments for these neurocognitive disorders. However, cholinergic deep brain stimulation targeting the nucleus basalis of Meynert, a major source of cholinergic input to the cortex, has shown promise for improving cognitive function, specifically visual working memory and attention, in nonhuman primate models. Continuous stimulation at frequencies like those used in movement disorders impaired performance. However, intermittent stimulation in both adult and aged monkeys improved performance on delayed match-to-sample tasks. Additionally, these effects are accompanied by changes in neuronal activity in the prefrontal cortex, including increased firing rates, decreased trial-to-trial variability, broadened tuning, and changes in rhythmicity. Clinical studies in humans have demonstrated that DBS targeting the NB is well-tolerated and there is room for cognitive function improvement. Still, more research is needed to determine the optimal parameters of stimulation in humans and to fully elucidate the potential of cholinergic DBS as a therapeutic intervention for cognitive deficits. Overall, these findings highlight the importance of cholinergic system activation in cognitive function and suggest DBS of the NB may serve as an alternative therapy for treating neurocognitive disorders.

## Figures and Tables

**Figure 1 brainsci-13-00917-f001:**
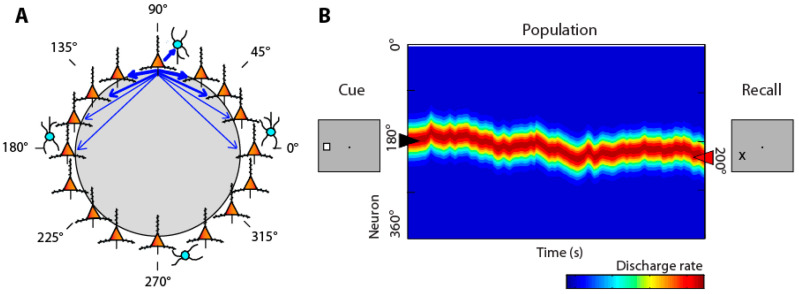
(**A**) Schematic diagram of the bump attractor model. Neurons representing different stimulus location are arranged in a ring so as to indicate each neuron’s preferred location. Synaptic connections between neurons with similar preference are stronger. (**B**) Schematic of the pattern of activity in the network. Y axis represents neurons in the ring with different spatial preference. A stimulus appearing at the left creates a peak of activity in the network centered on the 180° location. This peak may drift in time during the delay period of the task. The location of the peak at the end of the delay period determines the location that the subject recalls. From Ref. [55].

**Figure 2 brainsci-13-00917-f002:**
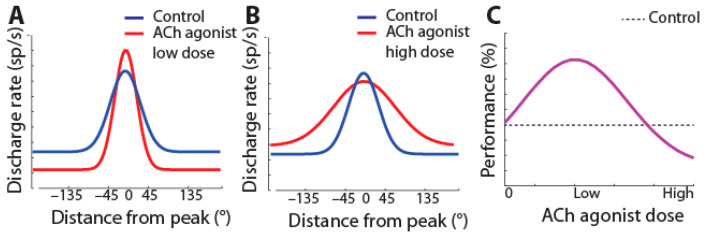
(**A**) Schematic effect of iontophoresis of low doses of cholinergic agonists on tuning of prefrontal neurons: firing rate increases at the neuron’s preferred (peak) location, resulting in sharpened tuning. (**B**) At high doses, agonists broaden tuning. (**C**) Hypothesized effect of dosage of cholinergic agonists on performance: improvement of performance is expected at doses that sharpen tuning.

**Figure 3 brainsci-13-00917-f003:**
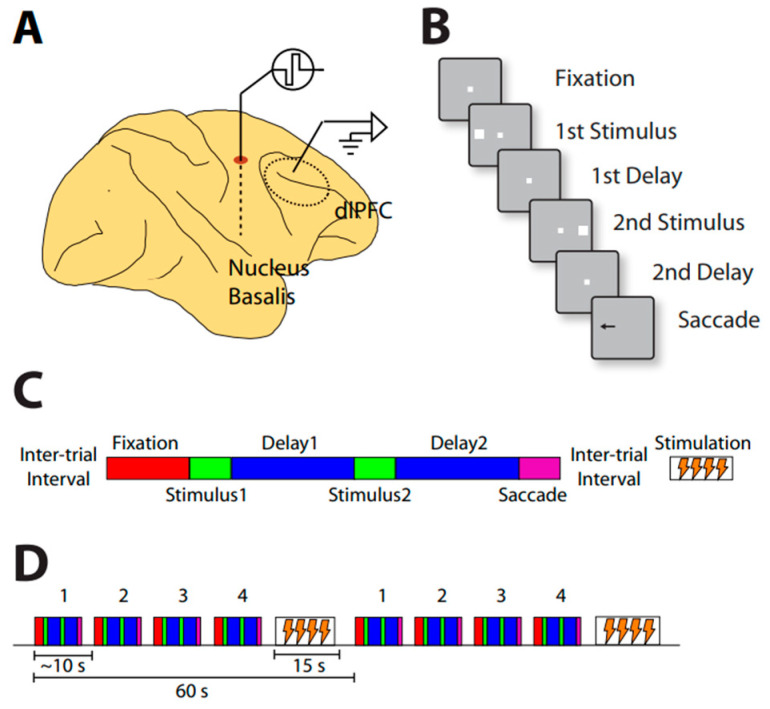
(**A**) Illustration of simultaneous NB stimulation and neurophysiological recordings from the prefrontal cortex. (**B**) Schematic diagram of the behavioral task. Successive frames represent task events. (**C**) Diagram of the relative timing of task events and NB stimulation. Behavioral trials comprise the fixation period, visual stimulus presentation, delay period, and response periods, now represented by different color rectangles. Stimulation is delivered in between trials after the saccade. (**D**) Trial structure during neural recording. Behavioral trials lasting approximately 45 s are interleaved with 15 s periods of stimulation, when no behavior is performed. Comparisons can be made between blocks of trials involving stimulation during the 15 s pause period, and blocks of trials not delivering stimulation (sham).

**Figure 4 brainsci-13-00917-f004:**
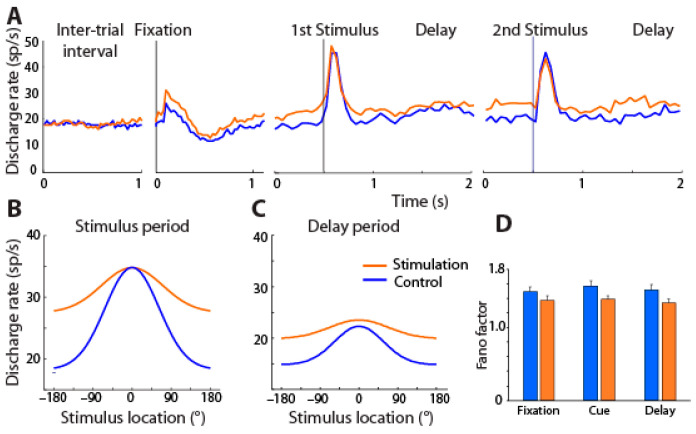
(**A**) Mean firing rate across PFC neurons with (orange) and without NB stimulation (blue) is shown for different task intervals, involving presentation of the best visual stimulus of each neuron. (**B**,**C**) Average tuning function from all PFC neurons for cue and delay period with and without stimulation; best stimulus has been shifted to 0° location in each case. (**D**) Mean Fano factor of spike counts during the fixation, cue, and delay period of the task under control and NB stimulation. From Ref. [98].

**Table 1 brainsci-13-00917-t001:** Human clinical studies targeting the nucleus basalis of Meynert (NB) [76,113,114,115,116,117,118,119,120].

Study First Author	Year of Publication	Type of Patients	Target	Number of Patients
Freund et al. [76]	2009	Parkinson’s–dementia	STN and NB	1
Kuhn et al. [113]	2015	Alzheimer’s	NB	6
Gratwicke et al. [114]	2017	Parkinson’s	NB	6
Nombela et al. [116]	2019	Parkinson’s–dementia	GPi and NB	1
Gratwicke et al. [120]	2020	Lewy body dementia	NB	6
Maltête et al. [117]	2021	Lewy body dementia	NB	6
Sasikumar et al. [115]	2022	Parkinson’s	GPi and NB	6
Jiang et al. [118]	2022	Alzheimer’s	NB	8
Bogdan et al. [119]	2022	Parkinson’s	GPi and NB	12

## Data Availability

No new data were created for this study. All data relevant to this research can be found in the original studies cited.

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
