# Peer review of "Improving Visual Working Memory with Cholinergic Deep Brain Stimulation"

_brainsci, 2023, doi:10.3390/brainsci13060917_

Round 1

Reviewer 1 Report

The manuscript presented from Janki M Bava et al is interesting and original. However there are some critical point:

- The authors should extend the introduction concerning the role of neurotransmitters.

- The authors in the text mentioned that "Cholinergic agonists generally increase the activity of prefrontal neurons", but in this context they should improve the text explaining how, which are the molecular pathway involved etc.

- The authors should improve the discussion and conclusion including a paragraph concerning the limitation of their approach.

Author Response

1. We wish to thank the reviewers for their constructive comments that allowed us to improve the manuscript. We have now expanded the introduction with a more detailed review of the role of neurotransmitter systems on cognitive functions,.
2. We have also extended the section on cholinergic agonists, and provided an outline of molecular pathways involved.
3. Finally we have added a paragraph in the last section of the discussion, before the conclusions, expanding on current limitations of the approach.

Reviewer 2 Report

1.     The abbreviation should be completely described in the first appearance.

2.     Could the authors provide a table of studies targeting the Nucleus Basalis of Meynert? It would be interesting to have a description of the adverse events. A study comparing adverse events showing a comparison with movement disorders should be provided (L 245 - 246).

Author Response

1. We now define all abbreviations the first time they are encountered in the body of the text.

2. We have added a table with all studies targeting the Nucleus Basalis of Meynert. Considering the very small sample of patients tested so far, a systematic comparison with movement disorders (which have involved thousands of patients) is not possible, at present. We outline adverse events in the final paragraph of the discussion, prior to conclusions (as reviewer 1 requested)

Reviewer 3 Report

This topic is interesting, but this paper is not a systematic review. I have several concerns about this paper.

- In the abstract, the authors talked about the "nucleus basalis of Meynert", but they did not discuss about it in "8. Potential for Human Intervention" at the end of the manuscript. and in the conclusion section.

- In the introduction section, it is not clear what is the purpose of this paper. Revise it.

- Paper does not have a PRISMA flowchart in a methods section. How these papers were selected?

- "6. Deep Brain Stimulation" what do they add new? references are usually old. look at more recent: -- doi: 10.3390/ijerph19148799 -- doi: 10.1016/j.wneu.2022.04.084

- lines 262-263. Sentences like "Continuous stimulation at frequencies like those used in movement disorders impaired performance" does not make sense. Revise it.

- Figures 1, 3 and 4 are taken from previous papers. Authors should add new figures of the discussed topic.

This topic is interesting, but this paper is not a systematic review. I have several concerns about this paper.  Paper does not have a PRISMA flowchart in a methods section. How these papers were selected? What is the purpose of this paper? this it not clear.

Minor editing of English language required.

Author Response

1. We now discuss interventions targeting the nucleus Basalis in sections 8 and 9.
2. We have revised the introduction and provided more thorough background on the topic and motivation for our study.
3. Our goal was not to provide a systematic review of a topic that involves a large literature and for which a flowchart for the selection of studies would be essential. Instead, we wish to highlight a few recent developments, drawing from the broader literature.
4. We appreciated the reviewer’s suggestions. We now cite these articles.
5. We have revised accordingly lines 262-263 of the text (now page 6 – lines 239-240).
6. We have redesigned figure 3 and modified figure 4 to highlight our point better. We still use the original
figure 1 from a previous article because it provides essential background to readers.

Reviewer 4 Report

The following review article entitled ''Improving Visual Working Memory with Cholinergic Deep Brain Stimulation” brings relevant data in the context of an interesting issue. Generally, the manuscript is well written. The ideas are clearly presented. There are additional concerns need to be discussed. Please consider the following comments.

l   The background introduction of the study is not sufficiently detailed and need to be clarified.

l   Nonetheless, a weak point for the present study is a lack of description of the present hurdles, which represents a starting point in future investigations. The authors should further discuss the added value of this review in light of the existing literature.

I would appreciate a revised manuscript from the authors for further consideration.

minor copyediting for the language

Author Response

1. In response to several reviewers, we have revised the background section and expanded the introduction on neuromodulatory systems.
2. We now explain our motivation for composing this review and have also added a paragraph in the last section of the discussion, before the conclusions, expanding on current limitations of the approach.

Round 2

Reviewer 1 Report

The authors improved the quality of the manuscript.

Reviewer 3 Report

Authors solved all my criticisms.

Reviewer 4 Report

This manuscript can be accepted in the present revised form.